# Is birth out-of-hospital associated with mortality and morbidity by seven years of age?

Katja Ovaskainen[1,2]☯*, Riitta Ojala[1,3]☯, Mika Gissler[4,5]☯, Tiina Luukkaala[6,7]☯, Outi Tammela[1]☯

1 Department of Pediatrics, Tampere University Hospital, Tampere, Finland, 2 School of Medicine Doctoral Programme, University of Tampere, Tampere, Finland, 3 Center for Child Health Research, Tampere University and University Hospital, Tampere, Finland, 4 Finnish Institute for Health and Welfare, Helsinki, Finland, 5 Karolinska Institutet, Stockholm, Sweden, 6 Tampere University Hospital, Research, Development and Innovation Center, Tampere, Finland, 7 Tampere University, Faculty of Social Sciences, Health Sciences, Tampere, Finland

☯ These authors contributed equally to this work.
* katja.ovaskainen@gmail.com

## Abstract

### Background and aims

Compared to in-hospital births, the long-term outcome of children born out-of-hospital, planned or unplanned, is poorly studied. This study aimed to examine mortality and morbidity by seven years of age in children born out-of-hospital compared to those born in-hospital.

### Methods

This study was registered retrospectively and included 790 136 children born in Finland between 1996 and 2013. The study population was divided into three groups according to birth site: in-hospital (n = 788 622), planned out-of-hospital (n = 176), and unplanned out-of-hospital (n = 1338). Data regarding deaths, hospital visits, reimbursement of medical expenses, and disability allowances was collected up to seven years of age or by the year-end of 2018. The association between birth site and childhood morbidity was determined using multivariable-adjusted Cox hazard regression analysis.

### Results

No deaths were reported during the first seven years after birth in the children born out-of-hospital. The percentage of children with hospital visits due to infection by seven years of age was lower in those born planned out-of-hospital and in the combined planned out-of-hospital and unplanned out-of-hospital group compared to those born in-hospital. Furthermore, the percentage of children with hospital visits and who received disability allowances due to neurological or mental disorders was higher among those born unplanned out-of-hospital and out-of-hospital in total when compared to those born in-hospital. In the multivariable-adjusted Cox proportional hazard regression analysis, the hazard ratio for hospital visits due to asthma and/or allergic diseases (HR 0.84; 95% CI 0.72–0.98) was lower in

**Data Availability Statement:** Relevant data are within the manuscript and its tables. The original register data with identification codes are confidential and available only for the study group and the holding register worker. The original data

received from register holders cannot be shared publicly; however, researchers who meet the criteria for access to confidential data may apply for data access from Findata, the Health and Social Data Permit Authority (https://www.findata.fi/en/).

**Funding:** The Ministry of Social Affairs and Health grants research funding to the research committees of the ERVAs of all university hospitals. One of the authors and the supervisor of my thesis, docent Outi Tammela, received this funding for her study group (grant number 9X054). With the help of this funding I received salary and I was able work with this study as a full-time researcher.

**Competing interests:** The authors have declared that no competing interests exist.

children born out-of-hospital when compared to those born in-hospital. A similar decreased risk was found due to infections (HR 0.76; 95% CI 0.68–0.84). However, the risk for neurological or mental health disorders was similar between the children born in-hospital and out-of-hospital.

## Conclusions

Morbidity related to asthma or allergic diseases and infections by seven years of age appeared to be lower in children born out-of-hospital. Birth out-of-hospital seemed to not be associated with increased risk for neurological morbidity nor early childhood mortality. Our study groups were small and heterogeneous and because of this the results need to be interpreted with caution.

## Introduction

Compared to in-hospital births, the possible long-term risks and outcomes for children delivered planned out-of-hospital (POHD) and unplanned out-of-hospital (UOHD) are poorly known. Previous studies on the possible risks of out-of-hospital deliveries (OHDs) for mothers and offspring are limited almost completely to the perinatal or neonatal period. In our previous studies, we analyzed register-based data on POHDs and UOHDs during the perinatal period [1–3]. We found that perinatal mortality and morbidity rates were significantly higher among UOHD infants compared to children born in-hospital [3]. Such differences were less significant when we compared POHDs to in-hospital deliveries [1]. There are only a limited number of studies establishing the long-term outcomes of infants born out-of-hospital. A recently published study [4] showed that hospitalization rates were lower in UOHD children compared to those born in-hospital. Moreover, an Australian population-based cohort study showed that very preterm UOHD infants were significantly more likely to die within 28 days or one year of age compared to those born in-hospital. Furthermore, the study showed that only 41% of very preterm UOHD infants were alive at one year of age [5]. To our knowledge, these are the only studies related to this issue.

This study aimed to examine the association between neonatal, infant, and childhood mortality and morbidity between children delivered out-of-hospital and those delivered in-hospital.

## Material and methods

This national retrospective register study included all births in Finland between 1996 and 2013 registered in the Medical Birth Register, which contains data related to all live births and stillbirths from the gestational age of 22+0 weeks onward or those with a birth weight of at least 500 g. In addition, data in the Medical Birth Register includes maternal and delivery characteristics and obstetric procedures. During the study years, 1 053 802 infants were born in Finland. Infants who were stillborn or died before the age of seven days (n = 5322) were excluded in order to avoid overlapping with our previous reports [1, 3]. Infants with missing data regarding birth site (n = 1046), and at least one major congenital anomaly as well as operative deliveries (n = 257 296) were also excluded. Operative deliveries were excluded as a potential confounder in the analysis due to lack of operative deliveries in the out-of-hospital birth group. Overall, 790 136 children born in Finland between 1996 and 2013 were assessed.

The data from the Medical Birth Register was linked to the Care Register for Health Care and the Register of Congenital Malformations maintained by the Finnish Institute for Health and Welfare; to health insurance data from the Social Insurance Institution; and to the Cause-of-Death Register, maintained by Statistics Finland.

Morbidity data was collected from the Care Register for Health Care, including data on inpatient visits, admissions to public hospitals, specialized health care outpatient visits (available since 1998), and admission and discharge dates. Primary health care visits were not included. Data regarding reimbursements of medicine expenses and disability allowances was obtained from the Social Insurance Institution. In Finland, reimbursements of medicine expenses and disability allowances can be granted for children if they require long-term therapy for a disease. The follow-up time was up to seven years of age or by the year-end of 2018.

Finnish growth curves were used to analyze birth weight in relation to gestational age [6]. Small for gestational age (SGA) was defined as a birth weight two or more standard deviations (SDs) below the population average. Large for gestational age (LGA) was defined as a birth weight two or more SDs above the average. Socioeconomic status was defined by using the mother's occupation and divided into four groups: upper-level employees, lower-level employees, manual workers, and others. "Others" included students, housewives, and unclassifiable cases. Congenital anomalies were divided into major and minor anomalies according to the classification of the European surveillance of congenital anomalies [7]. Only major congenital anomalies were reported because the reporting of minor anomalies varies depending on time period and the reporting unit. The Medical Birth Register includes data regarding Apgar scores at one and five minutes, but data on five- minute Apgar scores has only been available since 2004. Five-minute Apgar scores were not analyzed in our study due to missing data including all information from the first eight years of our study period.

For the analysis of morbidity, three main groups based on certain ICD-10 codes, Anatomical Therapeutic Chemical (ATC) classification codes, and reimbursement codes were created: asthma and allergy (ICD-10 codes: J45* asthma, L20.0 atopic dermatitis, L27.2 dermatitis due to ingested food, K52.2 allergic and dietetic gastroenteritis and colitis, J30.10 allergic rhinitis due to pollen, and J30.3 other allergic rhinitis; ATC-codes: R03 drugs for obstructive airway diseases and V06DF milk substitutes; and reimbursement codes: 203 chronic asthma and 505 and 506 milk allergy), central nervous system-associated problems, including neurological and mental health diseases and sensory impairments (ICD-10 codes: G40* epilepsy, G41* status epilepticus, G80*-G83* cerebral palsy and other paralytical syndromes, F90-98* behavioral and emotional disorders, F70-79* intellectual disabilities, F82, F83, F84* different developmental disorders, H90* hearing loss, and H54* blindness; ATC-codes: N03* drugs for epilepsy, and N06* antidepressants, psycholeptics, etc.; and reimbursement codes 111, 181, 182, 183, 199 epilepsy, 112 severe psychosis and other severe mental disorders, and 113 behavioral disorders with intellectual disabilities) and infections (ICD-10 codes: H65*-H66* otitis media, J06* upper respiratory infections, B34.9 unspecified viral infection, J12*-J18* pneumonia, and J20*-J21* acute bronchitis and bronchiolitis). Based on the data in the Care Register for Health Care and the Social Insurance Institution, we determined incidences for the previously mentioned morbidity groups. The age at diagnosis for certain morbidity groups was defined according to the first detection in one of the aforementioned registers. Hospital visits included all hospitalizations: inpatient visits at all hospitals and specialized health care outpatient visits at public hospitals.

Based on the age at the time of death, mortality after the perinatal period was divided into three categories: seven to 27 days, 28 days to 11 months and one year or older.

There was no patient or public involvement in setting the research question, designing the study, or interpreting the study results. The research project was approved on behalf of the

whole country of Finland by the Ethics Committee of the Tampere Region (Date: 8.1.2013, No: R12268).

## Statistical analyses

The characteristics of the infants in this study and reference groups and those of their mothers were described with medians with an interquartile range in skew distributions. Variables which were categorical were described with percentages. Logistic and multinomial logistic regression analyses were performed to investigate the factors of UOHDs, POHDs, and out-of-hospital deliveries (OHDs) in comparison to in-hospital deliveries. The results were shown via odds ratios with 95% confidence intervals (CIs). All variables were entered simultaneously into the multivariable-adjusted models. The associations between birth site and childhood morbidity were determined using Cox hazard regression models for hospital visits for asthma or allergy, infection and neurological or mental disorder by the age of seven years. Risk factors included in the multivariable-models were: maternal age, mother living in partnership, socio-economic status, smoking during pregnancy, primipara, sex, length of gestation at birth less than $37^{+0}$ weeks (preterm) vs. $37^{+0}$ weeks or more (term), appropriate for gestational age (AGA)/SGA/LGA, one- minute Apgar score, treatment in a neonatal care unit and antibiotic treatment. Statistically significant ($p < 0.05$) interactions between birth site and other risk factors were included into the final multivariable-adjusted model. Cox proportional assumptions were tested and found valid according to the site of birth. The results were expressed as hazard ratios (HRs) with 95% CIs. The analyses were carried out using IBM SPSS Statistics for Windows, Version 25.0 (IBM Corp., Armonk, NY).

## Results

The study population consisted of 176 POHD infants and 1338 UOHD infants. The reference group consisted of 788 622 infants born in-hospital. The characteristics of the mothers and infants are presented in Tables 1 and 2. Compared to the children born in-hospital, mothers' low socioeconomic status was less prevalent in the POHD group (4% vs. 14.6%), but the percentage of missing data was almost 20% for this variable. Nationality was more often other than Finnish in the POHD group (8.6% vs. 4.7%), and there were fewer primiparas among the POHD (11.4%) and UOHD mothers (9.6%) than among those who gave birth in-hospital (35.8%). Furthermore, mothers with POHDs smoked less often during pregnancy (6.3% vs. 14.9%). Post term ($> 42^{+0}$ weeks) births were more frequent in the POHD (1.7%) group, but less frequent in the UOHD group (0.5%) when compared to the in-hospital group (2.4%). The POHD infants had higher birth weights, while admissions to the neonatal unit (2.8% vs. 10.7%), need of ventilator treatment (0.6% vs. 1.2%), and treatment with antibiotics (1.1% vs. 6.1%) were less frequent when compared to infants born in-hospital.

### Mortality

No childhood deaths were registered after the perinatal period among the POHD or UOHD children. The group born in-hospital had 1024 deaths (1.3 deaths per 1000 births) during the study period. Overall, 119 infants (11.6%) died at the age of seven to 27 days, 406 (39.7%) at the age of 28 days to 11 months, and 499 (48.8%) at the age of one to seven years.

### Morbidity

**Asthma and allergic diseases.** The need for hospital visits, reimbursement of medications, and disability allowances appeared to be similar in all groups, regardless of birth site

**Table 1. Characteristics of mothers (n = 790 136).**

| Site of birth | Planned home (n = 176) | | Unplanned out-of-hospital (n = 1338) | | In-hospital (n = 788 622) | |
|---|---|---|---|---|---|---|
| | n | (%) | n | (%) | n | (%) |
| **Study period, years** | | | | | | |
| 1996–2001 | 42 | (23.9) | 187 | (14.0) | 265 792 | (33.7) |
| 2002–2007 | 59 | (33.5) | 427 | (31.9) | 257 896 | (32.7) |
| 2008–2013 | 75 | (42.6) | 724 | (54.1) | 264 934 | (33.6) |
| Area of Finland | | | | | | |
| Southern | 62 | (35.2) | 408 | (30.5) | 269 098 | (34.1) |
| Eastern | 11 | (6.3) | 213 | (15.9) | 111 408 | (14.1) |
| Northern | 17 | (9.7) | 302 | (22.6) | 124 570 | (15.8) |
| Western | 60 | (34.1) | 258 | (19.3) | 156 979 | (19.9) |
| Southwestern | 26 | (14.8) | 147 | (11.0) | 121 494 | (15.4) |
| Mothers' age, years | | | | | | |
| ≤ 19 | 0 | (< 0.1) | 29 | (2.2) | 14 927 | (1.9) |
| 20–34 | 120 | (68.2) | 1002 | (74.9) | 619 992 | (78.6) |
| ≥ 35 | 56 | (31.8) | 307 | (22.9) | 153 773 | (19.5) |
| Cohabitation | | | | | | |
| Yes | 159 | (90.3) | 1177 | (88.0) | 700 314 | (88.8) |
| No | 11 | (6.3) | 88 | (6.6) | 43 753 | (5.5) |
| Unknown | 6 | (3.4) | 73 | (5.5) | 44 555 | (5.6) |
| Socioeconomic status | | | | | | |
| Upper-level employee | 36 | (20.5) | 147 | (11.1) | 123 098 | (15.6) |
| Lower-level employee | 50 | (28.4) | 406 | (30.3) | 280 383 | (35.5) |
| Manual worker | 7 | (4.0) | 207 | (15.5) | 115 417 | (14.6) |
| Other | 54 | (30.7) | 277 | (20.7) | 138 499 | (17.5) |
| Unknown | 29 | (16.5) | 301 | (22.5) | 131 733 | (16.7) |
| Nationality | | | | | | |
| Finnish | 150 | (85.2) | 1047 | (78.3) | 673 340 | (85.4) |
| Other | 14 | (8.0) | 115 | (8.6) | 37 372 | (4.7) |
| Unknown | 12 | (6.8) | 176 | (13.2) | 77 910 | (9.9) |
| Smoking | | | | | | |
| No | 153 | (86.9) | 1033 | (77.2) | 651 891 | (82.7) |
| Yes | 11 | (6.3) | 214 | (16.0) | 117 778 | (14.9) |
| Unknown | 12 | (6.8) | 91 | (6.8) | 18 953 | (2.4) |
| Primipara | | | | | | |
| No | 155 | (88.1) | 1210 | (90.4) | 506 097 | (64.2) |
| Yes | 20 | (11.4) | 128 | (9.6) | 282 174 | (35.8) |
| Unknown | | | | | | |
| Length of gestation at birth, weeks+days | | | | | | |
| 22+0–31+6 | 0 | (< 0.1) | 15 | (1.1) | 2562 | (0.3) |
| 32+0–36+6 | 1 | (0.6) | 68 | (5.1) | 28 740 | (3.6) |
| 37+0–42+0 | 151 | (85.8) | 1205 | (90.1) | 737 361 | (93.4) |
| > 42+0 | 10 | (5.7) | 18 | (1.3) | 18 613 | (2.4) |
| Unknown | 14 | (8.0) | 32 | (2.4) | 1894 | (0.2) |

(Table 3). In the Cox regression analysis, the hazard ratios for admission and outpatient visits to a hospital related to asthma or allergy for either the POHD or UOHD group children did not significantly differ from those born in-hospital. However, when POHD and UOHD groups

**Table 2. Characteristics of infants alive at age of seven days (n = 790 136).**

| Site of birth | Planned home (n = 176) | | Unplanned out-of-hospital (n = 1338) | | In-hospital (n = 788 622) | |
|---|---|---|---|---|---|---|
| | n | (%) | n | (%) | n | (%) |
| **Infant** | | | | | | |
| Boys | 82 | (46.6) | 664 | (49.6) | 395 586 | (50.2) |
| Birth weight, grams | | | | | | |
| 2500–3999 | 130 | (73.9) | 1098 | (82.1) | 627 315 | (79.5) |
| 4000–5500 | 31 | (17.6) | 169 | (12.6) | 140 820 | (17.9) |
| 1500–2499 | 1 | (0.6) | 57 | (4.3) | 18 571 | (2.4) |
| 500–1499 | 0 | (< 0.1) | 12 | (0.9) | 1701 | (0.2) |
| Unknown | 14 | (8.0) | 2 | (0.1) | 215 | (< 0.1) |
| Gestational weight | | | | | | |
| SGA (small for gestational age) | 0 | (< 0.1) | 27 | (2.0) | 12 442 | (1.6) |
| AGA (appropriate) | 159 | (90.3) | 1259 | (94.1) | 755 431 | (95.8) |
| LGA (large) | 0 | (< 0.1) | 19 | (1.4) | 18 603 | (2.4) |
| Unknown | 17 | (9.7) | 33 | (2.5) | 2146 | (0.3) |
| Apgar score 1min 0–6 | 7 | (4.0) | 85 | (6.4) | 25 457 | (3.2) |
| Admission to neonatal unit | 5 | (2.8) | 157 | (11.7) | 57 246 | (7.3) |
| Invasive ventilation | 1 | (0.6) | 17 | (1.3) | 4193 | (0.5) |
| Resuscitation at birth | 0 | (< 0.1) | 5 | (0.4) | 2808 | (0.4) |
| Antibiotic therapy in the first week of life | 2 | (1.1) | 90 | (6.7) | 27 828 | (3.5) |

were combined into one group (OHD group), the children born out-of-hospital appeared to exhibit significantly lower hazard ratios for asthma and allergy than the children born in-hospital. No significant interactions were found (Table 6).

## Infections

The percentage of children with hospital visits due to infection was significantly lower in the POHD group and in the combined OHD group than in the born in-hospital group. Children in the POHD group had their first hospital visit due to pneumonia and bronchitis or bronchiolitis at a younger age compared to those born in-hospital (Table 4). In the Cox regression analysis, the risk of hospital visits due to infection by seven years of age was significantly lower among the children in the POHD group, in the UOHD group, and in the OHD group in total than among those born in-hospital. The decreased risk of infections in the children in the OHD group remained significant in the analysis with interactions (Table 6).

**Table 3. Morbidity related to asthma or allergy (n = 790 316).**

| Asthma or allergy (J45*, L20.0, L27.2, J30.10, J30.3, K52.2) | Site of birth | | | | | | | |
|---|---|---|---|---|---|---|---|---|
| | Planned home (n = 176) | | Unplanned out-of-hospital (n = 1338) | | In-hospital (n = 788 622) | | Out-of-hospital (planned or unplanned) (n = 1514) | |
| Hospital visits, n (%) | 16 | (9.1) | 156 | (11.7) | 98 817 | (12.5) | 172 | (11.4) |
| Number of hospital visits, median MD (InterQuartile Range IQR) | 4.5 | (2.25–17.25) | 3 | (1–7.75) | 3 | (2–8) | 3 | (1–8) |
| Age at 1st hospital visit in years, MD (IQR) | 1.1 | (0.6–2.0) | 1.3 | (0.7–3.1) | 1.5 | (0.7–3.2) | 1.3 | (0.7–3.0) |
| Reimbursement for medication, n (%) | 4 | (2.3) | 52 | (3.9) | 28 681 | (3.6) | 56 | (3.7) |
| Age at 1st reimbursement in years, MD (IQR) | 2.75 | (0.36–5.13) | 1.72 | (0.74–3.39) | 2.02 | (0.67–4.15) | 1.7 | (0.7–3.4) |
| Disability allowance due to, n (%) | 2 | (1.1) | 22 | (1.6) | 14 690 | (1.9) | 24 | (1.6) |

**Table 4. Morbidity related to infections (n = 790 316).**

| Infections (H65*_H66*, J06*, B34.9, J12*-J18*, J20*-J21*) | Planned home (n = 176) | | Unplanned out-of-hospital (n = 1338) | | In-hospital (n = 788 622) | | Out-of-hospital (planned or unplanned) (n = 1514) | |
|---|---|---|---|---|---|---|---|---|
| Hospital visits, n (%) | 38 | (21.6) | 439 | (32.8) | 273 958 | (34.7) | 477 | (31.5) |
| Number of hospital visits, median MD (Interquartile range IQR) | 2 | (1–3.25) | 2 | (1–3) | 2 | (1–3) | 2 | (1–3) |
| Age at 1st hospital visit in years, MD (IQR) | 1.1 | (0.5–2.1) | 1.5 | (0.6–2.6) | 1.4 | (0.7–2.7) | 1.4 | (0.6–2.5) |
| Pneumonia (J12*-J18*) | | | | | | | | |
| Hospital visits, n (%) | 8 | (4.5) | 57 | (4.3) | 30 530 | (3.9) | 65 | (4.3) |
| Number of hospital visits, MD (IQR) | 1 | (1–1.75) | 2 | (1–2) | 1 | (1–2) | n/a | n/a |
| Age at 1st hospital visit in years, MD (IQR) | 1.2 | (0.6–2.6) | 1.9 | (1.1–3.9) | 2.2 | (1.3–3.7) | 1.9 | (1.1–3.7) |
| Bronchitis or bronchiolitis (J20*-J21*) | | | | | | | | |
| Hospital visits, n (%) | 9 | (5.1) | 123 | (9.2) | 67 840 | (8.6) | 132 | (8.7) |
| Number of hospital visits, MD(IQR) | 2 | (1.5–3.5) | 2 | (1–3) | 2 | (1–3) | n/a | n/a |
| Age at 1st hospital visit in years, MD (IQR) | 0.3 | (0.1–0.6) | 0.9 | (0.3–1.8) | 1.0 | (0.4–1.9) | 0.7 | (0.3–1.6) |

**Neurological or mental health disorders.** The percentage of children with hospital visits and who received disability allowances due to neurological or mental health disorders was higher in the UOHD and OHD groups than in the in-hospital group (Table 5). In the Cox regression analysis, the HRs for admissions or outpatient visits to a hospital due to neurological or mental disorders by seven years of age increased in UOHD children and OHD children in total, compared to children born in-hospital. The statistical significance was, however, lost in the analysis with interactions (Table 6).

## Discussion

The POHD and UOHD groups differed significantly in terms of maternal, pregnancy and infant characteristics, which inevitably led to difficulties in comparing these groups to in-hospital deliveries. There is also a risk of selection bias in the analyses. Differences between these groups, i.e. more preterm deliveries, lower socioeconomic position, and smoking among

**Table 5. Morbidity related to neurological or mental health disorders (n = 790 316).**

| Neurological or mental health disorder (G40-G41, G80*-83*, F90-98*, F70-79*, F82, F83, F84, H90*, H54*) | Planned home (n = 176) | | Unplanned out-of-hospital (n = 1338) | | In-hospital (n = 788 622) | | Out-of-hospital (planned or unplanned) (n = 1514) | |
|---|---|---|---|---|---|---|---|---|
| Hospital visits, n (%) | 10 | (5.7) | 100 | (7.5) | 42 653 | (5.4) | 110 | (7.3) |
| Number of hospital visits, median MD (interquartile range IQR) | 2 | (1–22) | 4 | (1–10) | 3 | (1–10) | 4 | (1–10.5) |
| Age at 1st hospital visit in years, MD (IQR) | 5.8 | (1.0–6.6) | 4.9 | (3.3–6.0) | 5.0 | (3.0–6.0) | 4.9 | (3.3–6.0) |
| Reimbursement for medication n (%) | 1 | (0.6) | 4 | (0.3) | 3044 | (0.4) | 5 | (0.3) |
| Age at 1st reimbursement in years, MD (IQR) | 0.10 | | 3.70 | (0.84–5.87) | 3.42 | (1.41–5.29) | 1.6 | (0.4–5.9) |
| Disability allowance due to, n (%) | 2 | (1.1) | 60 | (4.5) | 17 952 | (2.3) | 62 | (4.1) |

**Table 6. Cox hazard regression models regarding hospital visits for asthma or allergy, infection, and neurological or mental disorder by the age of 7 years in the children born in-hospital, planned out-of-hospital, and unplanned out-of-hospital.**

| | | | | | | Planned and/or unplanned out-of-hospital births vs. births in-hospital | | | | | |
| --- | --- | --- | --- | --- | --- | --- | --- | --- | --- | --- | --- |
| | | | | | | Univariable | | Multivariable without interactions | | Multivariable with interactions | |
| | **N** | **n** | **(%)** | **Pyrs** | **Risk** | **(95 % CI)** | **HR** | **(95 % CI)** | **HR** | **(95 % CI)** | **HR** | **(95 % CI)** |
| **Asthma or allergy** | **790 136** | **98 989** | **(12.5)** | **5 048 310** | **196** | **(195–197)** | | | | | | |
| In-hospital | 788 622 | 98 817 | (12) | 5 038 578 | 196 | (195–197) | 1.00 | | 1.00 | | 1.00 | |
| Planned out-of-hospital | 176 | 16 | (9) | 1141 | 140 | (72–208) | 0.72 | (0.44–1.17) | 0.71 | (0.44–1.17) | 0.63 | (0.35–1.10 |
| Unplanned out-of-hospital | 1338 | 156 | (12) | 8592 | 182 | (153–210) | 0.93 | (0.79–1.09) | 0.86 | (0.73–1.004) | 0.83 | (0.69–0.996) |
| In-hospital | 788 622 | 98 817 | (12.5) | 5 038 578 | 196 | (195–197) | 1.00 | | 1.00 | | 1.00 | |
| Out-of-hospital | 1514 | 172 | (11.4) | 9733 | 177 | (151–203) | 0.90 | (0.78–1.05) | 0.84 | (0.72–0.98) | 0.80 | (0.67–0.96) |
| **Infections** | **790 136** | **274 435** | **(34.7)** | **5 414 948** | **507** | **(505–509)** | | | | | | |
| In-hospital | 788 622 | 273 958 | (34.7) | 5 404 641 | 507 | (505–509) | 1.00 | | 1.00 | | 1.00 | |
| Planned out-of-hospital | 176 | 38 | (21.6) | 1207 | 315 | (216–413) | 0.59 | (0.43–0.80) | 0.59 | (0.43–0.81) | 0.49 | (0.34–0.72) |
| Unplanned out-of-hospital | 1338 | 439 | (32.8) | 9101 | 482 | (438–526) | 0.94 | (0.85–1.03) | 0.87 | (0.79–0.96) | 0.80 | (0.71–0.89) |
| In-hospital | 788 622 | 273 958 | (34.7) | 5 404 641 | 507 | (505–509) | 1.00 | | 1.00 | | 1.00 | |
| Out-of-hospital | 1,514 | 477 | (31.5) | 10 307 | 463 | (422–503) | 0.89 | (0.81–0.98) | 0.84 | (0.76–0.92) | 0.76 | (0.68–0.84) |
| **Neurological or mental disorder** | **790 136** | **42 763** | **(5.4)** | **4 134 687** | **103** | **(102–104)** | | | | | | |
| In-hospital | 788 622 | 42 653 | (5.4) | 4 126 554 | 103 | (102–104) | 1.00 | | 1.00 | | 1.00 | |
| Planned out-of-hospital | 176 | 10 | (5.7) | 1028 | 97 | (37–157) | 1.05 | (0.56–1.95) | 1.11 | (0.60–2.07) | 1.19 | (0.62–2.29) |
| Unplanned out-of-hospital | 1338 | 100 | (7.5) | 7106 | 141 | (112–168) | 1.40 | (1.15–1.70) | 1.24 | (1.01–1.51) | 1.04 | (0.83–1.31) |
| In-hospital | 722 622 | 42 653 | (5.4) | 4 126 554 | 103 | (102–104) | 1.00 | | 1.00 | | 1.00 | |
| Out-of-hospital | 1514 | 110 | (7.3) | 8134 | 135 | (110–160) | 1.36 | (1.12–1.64) | 1.22 | (1.01–1.48) | 1.06 | (0.85–1.31) |

Cox hazard regression models regarding hospital admissions for asthma or allergy, infection and neurological or mental disorder by the age of seven years were shown by person-years until seven age-years (Pyrs), risk per 10,000 person-years (Risk) and Cox proportional hazard regression estimates (HR) with 95% confidence intervals (CI). Statistically significant (p<0.05) interactions between the site of birth with other risk factors were included into the final multivariable-adjusted model.

UOHDs may have had an association with the risk of morbidities we studied. Thus, we realize that our results need to be interpreted with caution. Our study groups were small due to low out-of-hospital delivery rates in Finland, indicating that statistical significance may be difficult to show.

The findings of our study showed that the percentage of children with hospital visits due to infection by seven years of age was lower in those born planned and out-of-hospital in total than those born in-hospital. The percentage of children who had hospital visits and received disability allowances due to neurological or mental disorders was higher in those born unplanned out-of-hospital and out-of-hospital in total than in those born in-hospital. In the Cox regression analysis corrected with interactions the hazard ratios for visits to a hospital due to asthma or allergic diseases or due to infection were lower among the children born out-of-hospital.

The POHD group had the lowest percentage of children who needed hospital visits due to infection by seven years of age. Mothers who deliver at home as planned are more often older [8–18], non-smokers [12, 18–20], and married [14, 17, 21]. In addition, socioeconomic status and/or education are usually higher among these women [10, 12, 14, 17, 22, 23]. In our population, the POHD group children were of higher birth weights, rarely needed assisted ventilation, and had fewer admissions to the neonatal unit. Thus, when perinatal, neonatal, demographic, and socioeconomic factors are combined, it might result in conditions that make these children in the POHD group less prone to infections requiring hospital care.

Previous studies have shown that mothers who give birth unplanned out-of-hospital tend to be younger [5, 17, 24] or older [24–27], unmarried/not cohabiting [17, 24, 25], smoke during pregnancy [17, 25], have less education or a lower socioeconomic status [17, 27, 28], more likely substance abusers [29], not visit or have fewer prenatal care visits [4, 17, 30–34], and exhibit a lower length of gestation at delivery [5, 24, 25, 28, 30, 35]. These are obviously risk factors for requiring future hospital care and disability allowances due to neurodevelopmental problems.

One study previously reported on the long-term morbidity of 3580 children born unplanned out-of-hospital in a single tertiary hospital area in Israel [4]. The study population included 243 682 singleton deliveries. The hospitalization rates of the UOHD children by 18 years of age due to respiratory, infectious, and neurological causes were lower when compared to children born in-hospital. The author suggested that socioeconomic and demographic factors related to UOHDs might also be related to the under-utilization of health care services. Such under-utilization may be an unlikely phenomenon, and avoiding/reluctance toward health care visits is probably rare in the Finnish free public health insurance and uniform social security system.

Children born out-of-hospital in total is a very heterogeneous group in terms of perinatal, demographic, and socioeconomic factors. In most cases, the POHD group children remained in a home environment during their perinatal period. In contrast, the UOHD group children were mostly transported after birth to the hospital with their mothers, and some were even admitted to the neonatal unit. The only factor in common among this population was that these children were not born in a delivery room environment. A quite significant percentage of mothers delivering in-hospital receive intrapartum antibiotics. An American study reported that 38.3% of mothers received antibiotics for reasons such as group B Streptococcal (GBS)-positivity, suspected maternal infection, cesarean section, preterm labor, or prolonged membrane rupture [36]. According to unpublished data from the Finnish Medical Birth Register, 5.1% of women received intrapartum antibiotic prophylaxis during vaginal delivery to prevent GBS disease in their infants (years 2017–2019, excluding Southern Finland). In addition, women receive intrapartum antibiotic treatment, but number of these women is not studied or published in Finland. Instead, intrapartum exposure to antibiotics is lacking in all out-of-hospital deliveries. The association between the environment and possible intrapartum exposure to antibiotics at birth might have an impact on the children's skin and gut microbiome [37]. These above-mentioned factors may have protective or harmful effects on the prevalence of allergic and infectious diseases during the childhood.

The main strength of this study is the reliable data obtained from the Finnish national registers [38–41]. We included all OHD infants alive at the age of seven days—except those born after operative delivery or with major malformations.

One of the limitations of this retrospective register study is that we were only able to collect data for a limited range of variables available from the registers. This is why we were unable to analyze certain parameters, such as the prevalence of breastfeeding or smoking after pregnancy. We were also unable to gather information on whether the mother received antibiotic treatments during her pregnancy or delivery or whether inheritable risk factors, such as parental asthma, existed. Because of our study design, infants born at the end of the study period had shorter follow-up times than those born in earlier years. However, even the youngest children born toward the end of the study period were aged at least 5 years at the end of the period when the data was collected.

Future studies are indicated, in order to obtain more information on whether children born planned or unplanned out-of-hospital need additional follow-up during their childhood and whether there is increased risk for certain morbidities. This would be important in order

to set a proper follow-up for such children and share this information with families who are planning for a home birth.

## Conclusions

The risk of childhood morbidity related to asthma and allergic diseases, and infections seemed to be smaller among children born out-of-hospital. A reason for this could not be detected in our study, but is presumably multifactorial, including a possible lack of intrapartum antibiotic prophylaxis or treatment for mothers. Furthermore, birth out-of-hospital seemed to not be associated with childhood mortality. Our study groups were small and heterogeneous and because of this the results need to be interpreted with caution.

## Author Contributions

**Data curation:** Katja Ovaskainen.

**Formal analysis:** Katja Ovaskainen, Tiina Luukkaala.

**Methodology:** Katja Ovaskainen, Mika Gissler, Tiina Luukkaala.

**Resources:** Katja Ovaskainen, Mika Gissler.

**Supervision:** Riitta Ojala, Outi Tammela.

**Visualization:** Tiina Luukkaala.

**Writing – original draft:** Katja Ovaskainen.

**Writing – review & editing:** Katja Ovaskainen, Riitta Ojala, Mika Gissler, Tiina Luukkaala, Outi Tammela.

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
