## [Decision Letter · Decision Letter 0]

6 Jan 2021

PONE-D-20-37796

Is birth out-of-hospital associated with mortality and morbidity by 7 years of age?

PLOS ONE

Dear Dr. Ovaskainen,

Thank you for submitting your manuscript to PLOS ONE. After careful consideration, we feel that it has merit but does not fully meet PLOS ONE’s publication criteria as it currently stands. Therefore, we invite you to submit a revised version of the manuscript that addresses the points raised during the review process.

Apart from addressing the issues raised by the two reviewers, please add a financial disclosure and an ethical statement, as well as information about the availability of the full data (the manuscript contains summary data only which do not allow replication of findings). In the abstract, add N of births in the three groups, effect estimates and their confidence intervals, and in the results section, also report in the text N of the different groups. Please separate clearly findings for UOH and POH, since these are (as reviewers stated) two distinct groups. In the discussion, line 228, I suggest to refer to "higher" rather than "better" SES/education. Please control also the references - Ref. no. 26 and 29 are identical.

We look forward to receiving your revised manuscript.

Kind regards,

Barbara Schumann, Ph.D.

Academic Editor

PLOS ONE

Journal Requirements:

2. Thank you for stating the following financial disclosure: 'NO'

3. Thank you for stating the following in your Competing Interests section: 'NO'

a. Please complete your Competing Interests statement to state any Competing Interests. If you have no competing interests, please state "The authors have declared that no competing interests exist.", as detailed online in our guide for authors at http://journals.plos.org/plosone/s/submit-now

Reviewers' comments:

Reviewer's Responses to Questions

**Comments to the Author**

1. Is the manuscript technically sound, and do the data support the conclusions?

Reviewer #1: Partly

Reviewer #2: Partly

2. Has the statistical analysis been performed appropriately and rigorously? 

Reviewer #1: No

Reviewer #2: Yes

3. Have the authors made all data underlying the findings in their manuscript fully available?

Reviewer #1: Yes

Reviewer #2: Yes

4. Is the manuscript presented in an intelligible fashion and written in standard English?

Reviewer #1: No

Reviewer #2: Yes

5. Review Comments to the Author

Reviewer #1: The manuscript PONE-D-20-37796 presents results from a cohort of 790 136 children, born in Finland between 1996 and 2013. The primary research question was whether place of birth (in-hospital or out-of-hospital) influenced mortality and morbidity risk. As data on longterm mortality and morbidity risk of children, born inside and outside hospital were rare, the results are interesting.

Although it is a huge data set, the study suffers from the surprisingly low proportion of out-of-hospital (OH) births. Only 1514 (0,2%) children were born OH. The authors do not provide any explanation as to why the proportion of OH deliveries is so low in Finland compared to other countries. Because of the small OH proportion, the main research question (place of birth and mortality) cannot be examined conclusively The only valid conclusion is that out-of-hospital births obviously do not increase the mortality risk. The small proportion of out-of-hospital births further limits the question on morbidity considerably, since the observed incident diseases were relatively rare, especially for the group of planned out-of-hospital (POH) births.

Besides the low number of cases in the OH group, for which the authors cannot be held responsible, my main criticisms of the manuscript is the poor description of the methodology and presentation of results.

In detail I have the following comments: see attachement

Reviewer #2: PLOS ONE

Is birth out-of-hospital associated with mortality and morbidity by 7 years of age?

Manuscript Number: PONE-D-20-37796

Article Type: Research Article

The authors of this paper describe the impact of the morbidity and mortality of children at age seven, born out-of-hospital (UOH) as compared to those born in-hospital. The study period is almost two decades including almost 800 thousand deliveries in Finland during that period. A strong study on the possible risk of giving birth at home is important. At the same time, the responsibilities in analysing and presenting the data is great. From that point of view, this paper is important and at the same time the demands for a clear analysis and discussion are high. Although the study describes interesting data from a whole country, there are issues that need to be better clarified in the manuscript. I urge the authors to better address several points, especially in the discussion chapter.

The main conclusion of the authors is “The risk for morbidity related to asthma or allergic diseases and infections by 7 years of age was decreased in children born out-of-hospital. Birth out-of-hospital was not associated with increased risk for neurological morbidity and early childhood mortality.” This is a really strong claim, possibly too strong for the results presented in the paper. The authors even say that “….. our study groups were small, indicating that statistical significances may be difficult to show. Thus, we realize that our results need to be interpreted with caution.” I urge the authors to implement this caution in their conclusion.

An important limitation to this study is in my mind the possible (likely?) selection bias. Parents who choose to have the baby borne at home do certainly have a different background as mentioned in the paper and as can be seen in the tables. These include older mothers, more often multipara, upper socioeconomic class, less smoking etc. Moreover, and quite important, all but one baby in the planned out of-hospital (POH) group were delivered at term. In my mind, these are important differences and may certainly influence the results. The authors may want to add a substantial discussion on this to their paper, addressing clearly these differences and a possible selection bias..

Another important point for discussion or explanation is the number of children born UOH. The total number of children born in Finland during the study period was more than one million – certainly a respectable number! Of those almost 800 thousand were included in the study; 788.622 born in-hospital and 176 planned out-of-hospital (POH). Despite the large numbers, the total number of 176 POH over a period of almost two decades is not very high. It reflects approximately 10 children per year and the number of POH to in-hospital deliveries is approximately 1:5.000. This is alluded to in the section on limitations, nevertheless, I find that the authors draw quite strong conclusions from these numbers. These conclusions also partly contradict their conclusions in their earlier publications.

Following are several remarks, both minor as well as major that the authors may want to allude to.

Probably the group of unplanned out-of-hospital deliveries (UOH) is a very different group as compared to planned out of-hospital (POH) deliveries. The authors chose to combine these two groups in some instances. That may very well be interesting, but I find it a little confusing.

In the results chapter in the abstract no numbers/percentages/statistical results are mentioned.

The authors may want to explain why infants who died before the age of seven days were excluded from the analysis. Major congenital anomalies, stillbirths etc were already excluded. Why was mortality before seven days also excluded (apart from the congenital diseases)?

Similarly, a rather large number of missing data on the site of birth is surprising in a country like Finland with high health care standard and excellent registration.

The data collected included “data on inpatient visits in public hospitals, specialized health care outpatient visits”. It is not clear to me what “… specialized health care outpatient visits” means. Is this general practitioners/health care centres or specialised service like paediatricians, psychiatrists etc? It is also sometimes unclear in the analysis if the data was only collected for hospital visits or also for general health care visits. The difference between hospital visit and hospital admission should be clarified. Was there possibly a different health-care seeking behaviour between the groups?

I found the socioeconomic status of “others” somewhat difficult, see comment on results below. The authors may want to explain this. Similarly, combining neurological disorders with behavioural disorders may be questioned – although I can very well see the reason for this. The authors may want to comment on that.

The study was approved by the Ethics Committee of the Tampere. Obviously, I am not familiar with the process of ethical approvals in Finland - but is ethical approval in Tampere sufficient for a study including the whole of Finland?

The tables are hard to read, quite large and in my version extending over several pages with the header only on the first page. The authors my consider making the tables easier to read, emphasising better what might be significant and having the legends more descriptive.

The first sentence in the results chapter confuses me “Compared to children born in-hospital, the percentage of mothers aged ≤ 19 years was higher in the UOHD group, and mothers aged ≥ 35 was higher in the group of planned home births.” If I read the table correctly, the percentage of mothers aged ≤ 19 years are 1.9% (n=29) in UOHD and in-hospital 2.2%, only a marginal (at best) difference!

When comparing socio-economic status, other or unknown accounts for 40-50% in all groups. That is quite high. The high percentage of unknown is also disturbing, the authors might want to explain that.

In table 3, one line reads “Reimbursement for, n (%)”. Is there something missing?

Differences in this table are not significant but an ad hoc observation is that POH are younger at first hospital visit, the number of visits is higher, but they are older when reimbursement is registered. This is intriguing.

Dose this only include hospital visits or also primary health care visits?

The discussion opens with the finding of less infections in the POH and OH groups whereas neurological and mental disorders are more common in the UOH and OH groups. As I mentioned earlier, including the UOH group is somewhat confusing as this probably reflects a different group in terms of maternal background and perinatal complications. The argument that perinatal, maternal, socioeconomic and demographic factors may explain (at least partly) this difference seems logical.

The comparison in this chapter to other countries is difficult, especially with the high standard of care in Finland in mind. The authors compare antibiotic prescription in the perinatal period between Finland and USA. It must be emphasised that these populations are very different. Antibiotic usage in high income countries has been to high (as has been described also in Finland), this problem is probably even more severe in USA than in Finland. In addition, Caesarean section is way more common in USA than in Finland, rendering this comparison of little help. The socioeconomic differences as well as the structure of the health care is also quite different between these countries. Therefore, these comparisons must be made with caution.

The discussion of the effect on the microbiome is interesting. This theory is becoming more popular – but it must be stressed that this is still a theory. Moreover, the information on perinatal antibiotic treatment in this study was not available.

Finally, as I mentioned before, I find the conclusions drawn by the authors quite strong. The authors mention that the “….. study groups were small, indicating that statistical significances may be difficult to show.” They subsequently add that “….. we realize that our results need to be interpreted with caution.” Subsequently they conclude with the strong statement that the “….risk of childhood morbidity related to asthma and allergic diseases and infection is smaller among children born OH”. I suggest they phrase this conclusion more carefully.

I also suggest they add to the discussion that a selection bias may certainly account for some of the changes observed as the mothers in the POH group were older, were more often multipara, of upper socioeconomic class, infants were delivered at term, they smoked less etc.

As mentioned in the beginning, a study on this topic is important. The authors should certainly be acknowledged for this extensive study. Analysing and discussing the data carries huge responsibility and must be interpreted with caution.

6. PLOS authors have the option to publish the peer review history of their article (what does this mean?). If published, this will include your full peer review and any attached files.

Reviewer #1: No

Reviewer #2: No

---

## [Author Response · Author response to Decision Letter 0]

13 Feb 2021

Reviewer #1 Comments to the Author:

Although it is a huge data set, the study suffers from the surprisingly low proportion of out-of-hospital (OH) births. Only 1514 (0,2%) children were born OH. The authors do not provide any explanation as to why the proportion of OH deliveries is so low in Finland compared to other countries. Because of the small OH proportion, the main research question (place of birth and mortality) cannot be examined conclusively The only valid conclusion is that out-of-hospital births obviously do not increase the mortality risk. The small proportion of out-of-hospital births further limits the question on morbidity considerably, since the observed incident diseases were relatively rare, especially for the group of planned out-of-hospital (POH) births. 

We agree that the proportion of children who were born out-of-hospital is low, indicating that the statistical significances may be difficult to show, and this is why the results of our study need to be interpreted with caution. We admit that this is a weakness, as commented in the Discussion. Our data collected from national register is on the other hand reliable and collected from the long study period. The place of birth is registered for each birth and double-checked in unclear situations.

Besides the low number of cases in the OH group, for which the authors cannot be held responsible, my main criticisms of the manuscript is the poor description of the methodology and presentation of results. 

We have revised the manuscript, in order to improve the presentation of the methods and results.

In detail I have the following comments: 

- general remarks:

o You use too many abbreviations, sometimes with the same content (e.g. UOH and UOHD). This hinders the reading of the manuscript.

We thank the reviewer for the feedback and we have simplified the use of abbreviations. 

- Abstract:

o The abstract should contain the main results, quantitatively presented as effect sizes with confidence intervals

Hazard ratios with confidence intervals have been included in revised version of the Abstract.

o Avoid repetition. The statement that “birth out-of-hospital was not associated with childhood mortality” similar appears twice in the Abstract.

This has been corrected in the Abstract.

- Methods:

o Please explain why you are excluding approximately 25% of the original population from the analysis.

We have clarified this in revised version. The reason is avoiding overlapping with our previous reports.

o Apgar score is not mentioned in the methods section but surprisingly appears for the first time in the results section 

Thank you very much for detecting of the error. This has been corrected in the Methods.

o The reasons for planned and unplanned births outside the hospital are so different that the construction of a combined group seems questionable. I would recommend analyzing the groups separately only. 

We have omitted combined data from the Tables 1 and 2, containing data on characteristics of the mothers and children. However, the children born out-of-hospital have also some clinical features in common, i.e. being not born in the delivery room environment and lack of exposure to intra partum antibiotics, as explained in the Discussion. This is why we decided to also combine these groups in some of the analyses. 

- Statistical analyses:

o I would recommend to omit the statistical tests (U-test, Chi-square-test) and p-values. Statistical tests are only useful in case of predefined hypothesis and a prior sample size calculation. (Please note the current discussion on p-values in the academic literature, for example: Amrhein, V., Greenland, S. & McShane, B. Scientists rise up against statistical significance. Nature 567, 305-307, doi:10.1038/d41586-019-00857-9 (2019)).

This is a very good comment. A revision is made, as requested.

o Regarding regression models, you should always outline how you formed these models, i.e., which is the dependent variable, which is the exposure variable, and which covariates and interaction terms are in the model? Regarding interaction – what significance level did you use to include interaction terms? What is the rationale to analyze interaction? 

The statistical methods have been explained more thoroughly.

- Results:

o In general, the description of the results in the text is poor. Please describe and interpret the main information of the tables in the text quantitatively. 

The results have been written more informatively.

o All tables: Please attach legends. The legends should explain the abbreviations used. The composition of the sample should be described in the methods section and not again in the legend. 

o Table 1 and 2: In tables 1 and 2, only the sample should be presented descriptively. Why are logistic models used here? These do not serve to answer the primary research question but overload the tables and hinder readability. I would recommend to delete the results of the logistic models. 

o Table 1: Length of gestation at birth: Please describe the rows clearly (22+0-31+6???)

We have revised these above mentioned things in tables 1 and 2.

o Table 3-5: As written above, you should avoid the p-values. I would recommend means, relative frequencies, and the respective 95% confidence intervals.

The p-values have been omitted.

o Table 6: Why for asthma the column with interaction is empty? What do the asterisk mean? What are the covariates in the multivariate models? Please list the covariates in the legend. Merely writing, “something was higher” or is “statistically significant” without mention the effect size is not enough. 

The numbers into the asthma column with interaction have been added.

o Regading neurological or mental disorders, you state that “the analysis with

interactions suggested that birth weight SGA or LGA was associated with increased HRs for later neurological and mental disorders”. However, this is not your research question. Furthermore, in table 6 I see only an effect for SGA (HR = 1,59) but not for LGA (HR = 1,0

The comment regarding our research question is very correct. The data on SGA, LGA and AGA is only confusing and it has been left out from the Table.

We have modified the Result section according to the above mentioned suggestions of reviewer. 

Reviewer #2 Comments to the Author

The authors of this paper describe the impact of the morbidity and mortality of children at age seven, born out-of-hospital (UOH) as compared to those born in-hospital. The study period is almost two decades including almost 800 thousand deliveries in Finland during that period. A strong study on the possible risk of giving birth at home is important. At the same time, the responsibilities in analysing and presenting the data is great. From that point of view, this paper is important and at the same time the demands for a clear analysis and discussion are high. Although the study describes interesting data from a whole country, there are issues that need to be better clarified in the manuscript. I urge the authors to better address several points, especially in the discussion chapter.

The main conclusion of the authors is “The risk for morbidity related to asthma or allergic diseases and infections by 7 years of age was decreased in children born out-of-hospital. Birth out-of-hospital was not associated with increased risk for neurological morbidity and early childhood mortality.” This is a really strong claim, possibly too strong for the results presented in the paper. The authors even say that “….. our study groups were small, indicating that statistical significances may be difficult to show. Thus, we realize that our results need to be interpreted with caution.” I urge the authors to implement this caution in their conclusion.

This is a very good comment. We have written the conclusions into a more cautious form.

An important limitation to this study is in my mind the possible (likely?) selection bias. Parents who choose to have the baby borne at home do certainly have a different background as mentioned in the paper and as can be seen in the tables. These include older mothers, more often multipara, upper socioeconomic class, less smoking etc. Moreover, and quite important, all but one baby in the planned out of-hospital (POH) group were delivered at term. In my mind, these are important differences and may certainly influence the results. The authors may want to add a substantial discussion on this to their paper, addressing clearly these differences and a possible selection bias.

The risk of selection bias is true, indeed. We have added this aspect into the discussion.

Another important point for discussion or explanation is the number of children born UOH. The total number of children born in Finland during the study period was more than one million – certainly a respectable number! Of those almost 800 thousand were included in the study; 788.622 born in-hospital and 176 planned out-of-hospital (POH). Despite the large numbers, the total number of 176 POH over a period of almost two decades is not very high. It reflects approximately 10 children per year and the number of POH to in-hospital deliveries is approximately 1:5.000. This is alluded to in the section on limitations, nevertheless, I find that the authors draw quite strong conclusions from these numbers. These conclusions also partly contradict their conclusions in their earlier publications.

The conclusions have been drawn more cautiously, as suggested. However, we do not agree that our report might contradict the conclusions in our earlier publications, because the present study is focused on the long-term outcome of children born out-of-hospital.

Following are several remarks, both minor as well as major that the authors may want to allude to.

Probably the group of unplanned out-of-hospital deliveries (UOH) is a very different group as compared to planned out of-hospital (POH) deliveries. The authors chose to combine these two groups in some instances. That may very well be interesting, but I find it a little confusing.

Please see the reply to Reviewer 1.

In the results chapter in the abstract no numbers/percentages/statistical results are mentioned.

More details on statistical results have been added into the Abstract.

The authors may want to explain why infants who died before the age of seven days were excluded from the analysis. Major congenital anomalies, stillbirths etc were already excluded. Why was mortality before seven days also excluded (apart from the congenital diseases)?

The exclusions were made in order to avoid overlapping with our previous publications. Our main interest was in childhood diagnoses which are not given at early neonatal period.

This has been clarified in the Methods.

Similarly, a rather large number of missing data on the site of birth is surprising in a country like Finland with high health care standard and excellent registration.

The number of missing data might look high. It is less than 1/1000 cases in the total study sample of 1 053 802 newborns, however.

The data collected included “data on inpatient visits in public hospitals, specialized health care outpatient visits”. It is not clear to me what “… specialized health care outpatient visits” means. Is this general practitioners/health care centres or specialised service like paediatricians, psychiatrists etc? It is also sometimes unclear in the analysis if the data was only collected for hospital visits or also for general health care visits. The difference between hospital visit and hospital admission should be clarified. Was there possibly a different health-care seeking behaviour between the groups?

We have clarified the terms used in the text. Specialized health care outpatient visits include only specialized health care. Visits at general practitioners and health care centers were not included in our study. The comment on possibility of different health-care seeking behavior between the groups is good. In the Discussion we have speculated that such possibility might not be high because of the free public health insurance and general social security system in Finland.

I found the socioeconomic status of “others” somewhat difficult, see comment on results below. The authors may want to explain this. Similarly, combining neurological disorders with behavioural disorders may be questioned – although I can very well see the reason for this. The authors may want to comment on that.

We have combined certain subgroups because of small numbers especially in certain diagnoses.

The study was approved by the Ethics Committee of the Tampere. Obviously, I am not familiar with the process of ethical approvals in Finland - but is ethical approval in Tampere sufficient for a study including the whole of Finland?

The ethical approval in Tampere is valid for whole Finland, which has been added into the text.

The tables are hard to read, quite large and in my version extending over several pages with the header only on the first page. The authors my consider making the tables easier to read, emphasising better what might be significant and having the legends more descriptive.

The tables and the legends have been clarified, as requested.

The first sentence in the results chapter confuses me “Compared to children born in-hospital, the percentage of mothers aged ≤ 19 years was higher in the UOHD group, and mothers aged ≥ 35 was higher in the group of planned home births.” If I read the table correctly, the percentage of mothers aged ≤ 19 years are 1.9% (n=29) in UOHD and in-hospital 2.2%, only a marginal (at best) difference!

We have corrected this, as requested.

When comparing socio-economic status, other or unknown accounts for 40-50% in all groups. That is quite high. The high percentage of unknown is also disturbing, the authors might want to explain that.

As mentioned in the Methods, group “others” includes students, housewives and unclassifiable cases. The data on education and occupation in the health care records are based on self-reporting. Socioeconomic status unknown include mostly of persons, who prefer not to disclose this information. In general it is difficult to define a socio-economic status for young women, since many of them are still in education or at home with child(ren). A clarification on this has been added into the Methods. 

In table 3, one line reads “Reimbursement for, n (%)”. Is there something missing?

Missing information has been fulfilled. Reimbursement for medication is correct expression.

Differences in this table are not significant but an ad hoc observation is that POH are younger at first hospital visit, the number of visits is higher, but they are older when reimbursement is registered. This is intriguing.

Dose this only include hospital visits or also primary health care visits?

Both inpatient and outpatient-hospital visits were included as mentioned in the Methods. 

The discussion opens with the finding of less infections in the POH and OH groups whereas neurological and mental disorders are more common in the UOH and OH groups. As I mentioned earlier, including the UOH group is somewhat confusing as this probably reflects a different group in terms of maternal background and perinatal complications. The argument that perinatal, maternal, socioeconomic and demographic factors may explain (at least partly) this difference seems logical.

The comparison in this chapter to other countries is difficult, especially with the high standard of care in Finland in mind. The authors compare antibiotic prescription in the perinatal period between Finland and USA. It must be emphasised that these populations are very different. Antibiotic usage in high income countries has been to high (as has been described also in Finland), this problem is probably even more severe in USA than in Finland. In addition, Caesarean section is way more common in USA than in Finland, rendering this comparison of little help. The socioeconomic differences as well as the structure of the health care is also quite different between these countries. Therefore, these comparisons must be made with caution.

Thank you very much for this comment. This led us to seek more information on the use of intra partum antibiotics in Finland. The discussion has been modified according to the new information.

The discussion of the effect on the microbiome is interesting. This theory is becoming more popular – but it must be stressed that this is still a theory. Moreover, the information on perinatal antibiotic treatment in this study was not available.

Yes, the speculation on microbiome is a theory, which has been added into the text. 

Finally, as I mentioned before, I find the conclusions drawn by the authors quite strong. The authors mention that the “….. study groups were small, indicating that statistical significances may be difficult to show.” They subsequently add that “….. we realize that our results need to be interpreted with caution.” Subsequently they conclude with the strong statement that the “….risk of childhood morbidity related to asthma and allergic diseases and infection is smaller among children born OH”. I suggest they phrase this conclusion more carefully.

The phrasing has been re-written into a more cautious form.

I also suggest they add to the discussion that a selection bias may certainly account for some of the changes observed as the mothers in the POH group were older, were more often multipara, of upper socioeconomic class, infants were delivered at term, they smoked less etc.

The selection bias has been discussed, as requested.

As mentioned in the beginning, a study on this topic is important. The authors should certainly be acknowledged for this extensive study. Analysing and discussing the data carries huge responsibility and must be interpreted with caution. 

Thank you very much for the comments.

---

## [Decision Letter · Decision Letter 1]

2 Mar 2021

PONE-D-20-37796R1

Is birth out-of-hospital associated with mortality and morbidity by 7 years of age?

PLOS ONE

Dear Dr. Ovaskainen,

Thank you for submitting your manuscript to PLOS ONE. After careful consideration, we feel that it has merit but does not fully meet PLOS ONE’s publication criteria as it currently stands. Therefore, we invite you to submit a revised version of the manuscript that addresses the points raised during the review process.

The revised version has considered most of the concerns by reviewers. However, we kindly ask you to address specifically the following issues:

1) While the first reviewer accepted the revised version, and the second reviewer asked for minor revisions, the issues raised reg. the comparability of the study groups, and the conclusions which can be made on this basis are quite fundamental. We ask you therefore to revise the paper by weakening statements, in particular general conclusions in the abstract, results, discussion and conclusions sections about health differences between the two groups born in / outside hospital. (E.g. instead of stating "Results showed that...", write "Results indicated that..." or "Differences appeared to be...").

2) Please check English language (or use a professional service as support), especially in the modified text, to improve its clarity.

3) Tables: make sure the title is above each table, and provide a legend for abbreviations and definitions. Table 6, please remove empty rows and columns.

4) In the submission form, question "Financial Disclosure": please state the funding agency (which is now partly named in the manuscript acknowledgements). Provide also a reply reg. competing interests.

5) In the submission form, you state "all data are available", but tables contain only summary statistics. Explain in the form, section "data availability" why the full dataset cannot be accessed.

We look forward to receiving your revised manuscript.

Kind regards,

Barbara Schumann, Ph.D.

Academic Editor

PLOS ONE

Journal Requirements:

Reviewers' comments:

Reviewer's Responses to Questions

**Comments to the Author**

1. If the authors have adequately addressed your comments raised in a previous round of review and you feel that this manuscript is now acceptable for publication, you may indicate that here to bypass the “Comments to the Author” section, enter your conflict of interest statement in the “Confidential to Editor” section, and submit your "Accept" recommendation.

Reviewer #1: All comments have been addressed

Reviewer #2: (No Response)

2. Is the manuscript technically sound, and do the data support the conclusions?

Reviewer #1: Yes

Reviewer #2: Partly

3. Has the statistical analysis been performed appropriately and rigorously? 

Reviewer #1: Yes

Reviewer #2: Yes

4. Have the authors made all data underlying the findings in their manuscript fully available?

Reviewer #1: Yes

Reviewer #2: Yes

5. Is the manuscript presented in an intelligible fashion and written in standard English?

Reviewer #1: Yes

Reviewer #2: Yes

6. Review Comments to the Author

Reviewer #1: The reviewer thanks the authors for their satisfactory revision of the manuscript and for answering the reviewer questions. Unfortunately, the version with the marked changes is obviously based on an intermediate version. This means that not all changes are marked compared to the original version, which makes it difficult to check the changes. However, the authors addressed all comments and made significant improvements to the manuscript. I have no further comments.

Reviewer #2: Is birth out-of-hospital associated with mortality and morbidity by 7 years of age?

Manuscript Number: PONE-D-20-37796

Article Type: Research Article

Revision 2

The authors have responded to the comments from the previous revision. As I mentioned in my review, the paper is interesting and carries a lot of important data. The responsibility of analysing the data correctly and draw accurate conclusion is high. Misunderstanding of the conclusions may have consequences.

The answers to my comments are in general good (sometimes short!) but in my opinion, some of the answers do need a little more considerations.

I still have one major comment and a few minor issues.

In my previous comments I pointed out the important problem of comparing the group of in-hospital deliveries to the POHD. These groups are obviously different and hard to compare. The in-hospital delivery group differed in many ways, including….

- Mothers were more often of lower socioeconomic status

- Mothers smoked more

- More often primipara

- More often premature deliveries

- More often <2500 grams

- More often small for gestational age

- More often needed NICU (prematurity?)

- More often needed antibiotics in the first week (due to prematurity/NICU, etc?)

- Etc.

These factors may certainly have influence of the diagnosis/occurrence of asthma and/or infections in childhood. Premature babies, needing NICU with mothers that smoke etc are definitely at higher risk for asthma than babies born á terme with no complications.

As alluded to in my earlier comments, that analysing and drawing conclusions in a study like this carries huge responsibility. From the conclusions drawn, one might think that if you want to safe your unborn baby from getting asthma or infections you should deliver at home. From your data, I totally disagree with that possible conclusion. In that case you would have to compare those born at home to a similar group born in hospital. This has not been done and the groups are not comparable in that sense. In my earlier comments, I urged the authors to be much more cautious in their conclusion. In the answer they claim to have done that (“We have written the conclusions into a more cautious form “). Still I find in the abstract conclusion the first (main?) conclusion “Morbidity related to asthma or allergic diseases and infections by 7 years of age was decreased in children born out-of-hospital.” Even though the authors have added to the end of this paragraph a sentence on the small and heterogenous group, I do disagree with this conclusion.

In the final conclusion of the manuscript the authors now claim the following: “The risk of childhood morbidity related to asthma and allergic diseases and infections seemed to be smaller among children born out-of-hospital. Reason for this could not be detected in our study, but is presumably multifactorial, including possibly lack of intrapartum antibiotic prophylaxis or treatment for mothers.” Although the authors are more careful here (“seem to be smaller” rather than “was decreased”) they attach the hypothesis that this might be due to lack of intrapartum antibiotics. This is not supported by their data and several other factors may play an important role as mentioned above (prematurity, smoking at home, SGA, NICU etc). Why is antibiotic treatment more likely to be the cause than all the other differences in the two groups?

I strongly encourage the authors to consider these comments and make adaptations to the text. I still maintain what I said in my earlier comments that the paper is interesting, and the study is large with important information. However, if the analysis and the main conclusions are questionable, that might decrease the credibility of this large and interesting study. Rather than adding only one sentence on selectin bias (“The POHD and UOHD groups differed also significantly in terms of maternal, pregnancy and infant characteristics, which inevitably leads to a risk of selection bias in the analyses.”) the authors may want to add a discussion on the clear and obvious differences between the two groups, possibly as the first paragraph in the discussion. Addressing this issue in a clear and open way would in my mind increase the credibility and importance of the study.

Minor comments:

In my earlier comments I mentioned that it was unclear to me why infants who died before the age of seven days were excluded from the analysis. The explanation in the reply is „) were excluded , in order to avoid overlapping with our previous reports).“ This is probably just fine – but did not help me understand why they were excluded until I read their earlier papers. A little inconvenient!

The discussion on specialized care, outpatient hospital visits and admissions has been clarified. In the same paragraph, the authors explain that different health-seeking behaviour is unlikely “because of the free public health insurance and general social security system in Finland.” I do understand this claim. On the other hand, if I understand It correctly, choosing a place of delivery and health-seeking behaviour in the Nordic countries is not primarily driven by economic perspective but rather philosophical and/or anthroposophical and/or sociological convictions as in some other European countries. Is it possible, that parents that choose to deliver at home have a different attitude towards hospitals and would subsequently not seek specialized care for their infant/baby? If that is possible, could it be that these parents would rather go to a general practitioner for some health issues like asthma and get treatment there? That might also skew the results for asthma diagnosis in the POHD group. Could this be region related (more than 80% of the POHD were in southern, western or southwestern part of Finland)?

The authors might want to address these points.

As I have repeatedly mentioned, the study is interesting. I would again like to encourage the authors to clearly address and discuss the clear differences between the groups so that incorrect conclusion could not be drawn (by others).

I am looking forward to reading their future studies on this issue.

7. PLOS authors have the option to publish the peer review history of their article (what does this mean?). If published, this will include your full peer review and any attached files.

Reviewer #1: No

Reviewer #2: No

---

## [Author Response · Author response to Decision Letter 1]

18 Mar 2021

The revised version has considered most of the concerns by reviewers. However, we kindly ask you to address specifically the following issues:

1) While the first reviewer accepted the revised version, and the second reviewer asked for minor revisions, the issues raised reg. the comparability of the study groups, and the conclusions which can be made on this basis are quite fundamental. We ask you therefore to revise the paper by weakening statements, in particular general conclusions in the abstract, results, discussion and conclusions sections about health differences between the two groups born in / outside hospital. (E.g. instead of stating "Results showed that...", write "Results indicated that..." or "Differences appeared to be...").

We have further weakened statements.

2) Please check English language (or use a professional service as support), especially in the modified text, to improve its clarity.

We have used a proofreading service for reviewing this latest revised manuscript.

3) Tables: make sure the title is above each table, and provide a legend for abbreviations and definitions. Table 6, please remove empty rows and columns.

Above mentioned thing have revised in tables.

4) In the submission form, question "Financial Disclosure": please state the funding agency (which is now partly named in the manuscript acknowledgements). Provide also a reply reg. competing interests.

This will be corrected in the submission form

5) In the submission form, you state "all data are available", but tables contain only summary statistics. Explain in the form, section "data availability" why the full dataset cannot be accessed.

All relevant data are within the manuscript and its tables. The original register data with identification codes is confidential and available only for the study group and the holding register worker. The original data received from register holders cannot be shared publicly. This will be added also in the submission form.

---

## [Editor Report · Decision Letter 2]

1 Apr 2021

Is birth out-of-hospital associated with mortality and morbidity by seven years of age?

PONE-D-20-37796R2

Dear Dr. Ovaskainen,

We’re pleased to inform you that your manuscript has been judged scientifically suitable for publication and will be formally accepted for publication once it meets all outstanding technical requirements.

Kind regards,

Barbara Schumann, Ph.D.

Academic Editor

PLOS ONE
---

## [Editor Report · Acceptance letter]

12 Apr 2021

PONE-D-20-37796R2 

Is birth out-of-hospital associated with mortality and morbidity by seven years of age? 

Dear Dr. Ovaskainen:

I'm pleased to inform you that your manuscript has been deemed suitable for publication in PLOS ONE. Congratulations! Your manuscript is now with our production department. 

Kind regards, 

on behalf of

Dr. Barbara Schumann 

Academic Editor

PLOS ONE